# Comparative Study of Different Walnut (*Juglans regia* L.) Varieties Based on Their Nutritional Values

**DOI:** 10.3390/plants13152097

**Published:** 2024-07-29

**Authors:** Lilla Szalóki-Dorkó, Pradeep Kumar, Dóra Székely, György Végvári, Gitta Ficzek, Gergely Simon, László Abrankó, Judit Tormási, Géza Bujdosó, Mónika Máté

**Affiliations:** 1Department of Fruit and Vegetable Processing Technology, Institute of Food Science and Technology, Hungarian University of Agriculture and Life Sciences, 29–43 Villányi út, 1118 Budapest, Hungary; szaloki-dorko.lilla@uni-mate.hu (L.S.-D.); pradiip06@gmail.com (P.K.); mate.monika.zsuzsanna@uni-mate.hu (M.M.); 2MHKSZ—Association of Hungarian Deepfreezing and Canning Industry, 2 Haller Utca, 1096 Budapest, Hungary; babinszki.szekely@gmail.com; 3Institute of Viticulture and Oenology, Eszterházy Károly Catholic University, 6–8 Leányka út, 3300 Eger, Hungary; vegvari.gyorgy@uni-eszterhazy.hu; 4Department of Fruit Growing, Institute of Horticultural Sciences, Hungarian University of Agriculture and Life Sciences, 29–43 Villányi út, 1118 Budapest, Hungary; ficzek.gitta@uni-mate.hu (G.F.); simon.gergely@uni-mate.hu (G.S.); 5Department of Food Chemistry and Analytical Chemistry, Institute of Food Science and Technology, Hungarian University of Agriculture and Life Sciences, 29–43 Villányi út, 1118 Budapest, Hungary; abranko.laszlo.peter@uni-mate.hu (L.A.); tormasi.judit@uni-mate.hu (J.T.); 6Institute for Horticultural Scienes, Fruit Growing Research Centre, Hungarian University of Agriculture and Life Science, 2 Park u., 1223 Budapest, Hungary

**Keywords:** *Juglans regia* L., fatty acid profile, polyphenols, mineral composition, fruit sites

## Abstract

Polyphenols, fatty acids, and mineral composition were studied in eight Persian walnut (*Juglans regia* L.) samples: ‘Milotai 10’ (M10) and ‘Alsószentiváni 117’ (A117) Hungarian varieties derived from two Hungarian growing areas (Pálháza—P and Berzék—B), ‘Chernivets’ky 1’ from Ukraine, and ‘Chandler’ from the United States of America, Chile, and Brazil purchased on the Hungarian consumer market. The aim was to reveal which walnut variety is the most suitable for consumption from a nutritive point of view. In comparison, both Hungarian varieties grown in Hungarian climatic conditions had on average approximately 33% more polyphenols, approximately 22% more SFAs (saturated fatty acids), and approximately 27% more MUFAs (monounsaturated fatty acids). Regarding the minerals, calcium, magnesium, phosphorus, and sodium were present in higher concentrations in both Hungarian varieties. Among the Hungarian-bred varieties, M10 mainly reached a higher compounds content, but the effects of the fruit site conditions were not clearly detected. Other varieties contained mainly potassium, copper, and zinc, such as ‘Chandler’ harvested from Chile. ‘Chernivets’ky 1’, harvested from Ukraine, had outstanding cinnamic acid and linoleic acid contents compared to the other examined varieties. All ‘Chandler’ samples (mainly the American) contained 11% more PUFAs compared to the samples derived from Hungary.

## 1. Introduction

Walnut (*Juglans regia* L.) is one of the traditional and widespread nut fruit species in temperate zones, especially in the Mediterranean and northern Mediterranean regions. Therefore, nuts are considered as a part of the “Mediterranean diet” with different health benefits [1]. Due to the numerous health benefits of the walnut, there is a keen interest in its production in many countries [2,3,4]. One of the greatest challenges of walnut production is the harmful effects of late-spring frosts [5,6], which can influence production. Other important issues are the harvest and the post-harvest stages, which can also influence the final kernel’s quality [5,7].

In recent years, nuts have become the focus of interest as healthy foods and their market is increasing continuously, which is due to their outstanding chemical composition. Walnuts are thus considered a ‘superfood’ due to their nutrient composition [8]. This nut is rich in polyphenols [9,10] such as flavonoids and phenolic acid [11], polyunsaturated fatty acids [12,13,14], minerals [15,16], essential amino acids [17], and bioactive peptides [18], and has a low glycemic index [19]. Currently, walnuts are not eaten just as a nut, but the bioactive compounds from them are incorporated into other food commodities to increase their nutritional value and health impacts. The consumption of walnuts is considered to be effective against several serious and lifestyle-related diseases. Several studies [20,21] showed that the consumption of walnuts in a moderate amount can reduce the levels of total cholesterol and positively alter the lipoprotein profile in men. The consumption of walnuts may also be beneficial against diabetes [22], cancer [23,24,25], age-related neurological disorders [26], or oxidative stress and inflammation [27].

To satisfy the increasing consumption of walnuts [28], the production of walnuts is also increasing worldwide. In the last two decades, China, the USA, Ukraine, Chile, and Turkey have been the leading walnut-producing countries. China (1.4 million metric tons of dried nuts) and the USA (682.2 million metric tons of dried nuts) are the largest producers [29]. Hungary produces 5950 T of dried walnuts every year [29], mostly for export to France, Germany, and Great Britain. The growing of Persian walnuts has a long history in Hungary, starting with the domestication of French varieties/genotypes, continuing with selective breeding from the local population. Currently, there are nine registered varieties on the National Variety List [30].

The aim of this research was to characterize and evaluate several commercially available walnut varieties on the Hungarian market, which is the first study in this area in Hungary. For this purpose, eight walnut varieties—including two Hungarian varieties, ‘Milotai 10’ (furthermore ‘M10’) and ‘Alsószentiváni 117’ (hereafter ‘A117’), from two growing areas in Hungary; ‘Chandler’ from America, Chile, and Brazil; and the Ukrainian variety ‘Chernivets’ky 1’—were compared with regard to their nutritional values. Our hypothesis is that fruit site conditions have some effects on the compounds and can be detected in the kernel. This work can be considered as a preliminary experiment, a pilot study on commercially available walnuts on the Hungarian market.

## 2. Results and Discussion

### 2.1. Polyphenols

Polyphenol content was determined using HPLC analysis (Section 3.2.1). The concentrations of cinnamic acid, gallic acid, rutin, and catechin are shown in Figure 1. For Hungarian varieties, rutin was present in the highest quantity among the examined polyphenols. The sample ‘M10’ from Pálháza contained 472.4 µg/g of rutin content followed by ‘M10’ from Berzék with a 427.5 µg/g value, while ‘A117’ had a lower concentration in both growing areas (348.9–367.7 µg/g). These data are approximately 70 and 60% higher than the results for the American Chandler variety. With regard to the four other walnuts, significantly (*p* = 0.01) lower rutin concentrations were measured in all samples compared to the Hungarian varieties. Ukrainian ‘Chernivets’ky 1’ contained 318.4 µg/g and Chilean ‘Chandler’ showed the lowest value (90.7 µg/g), which is almost a quarter of the rutin concentration in ‘M10’ varieties. Cinnamic acid was the second most abundant polyphenol component in the Hungarian walnuts, while the dominant component in the other walnuts, ‘A117’, had an almost 15% higher cinnamic acid concentration than ‘M10’ in both growing areas. The difference in concentration between the Hungarian and imported walnuts was not significant (*p* = 0.359). Brazil and American Chandler contained almost the same quantity, while Chilien Chandler was approximately 9% lower and ‘Chernivets’ky 1’ approximately 10% higher in cinnamic acid content. Gallic acid concentration was the highest in American ‘Chandler’ walnuts (183.28 µg/g) followed by the ‘M10’ sample from Berzék (117.35 µg/g), while walnuts exported from Brazil (41.45 µg/g) had the lowest content of gallic acid, which was approximately 78% lower compared to the control sample. Rutin and gallic acid contents of walnut samples in the study of Trandafir and Cosmulescu [31] were similar to our results. However, Wu et al. [10] detected only a 10.30 µg/g gallic acid concentration. Catechin was present only in ‘M10’ cultivars from both Hungarian growing areas in the examined walnuts in the range of 36.44–44.80 µg/g. Kafkas et al. [32] found catechin in 15 walnut samples grown in America and published a value in the range of 6.29–45.27 µg/g. Among the Hungarian varieties, ‘M10’ had higher gallic acid and rutin concentrations, while in case of cinnamic acid, ‘A117’ showed a higher content. However, the differences are not significant (*p* = 0.281; *p* = 0.063 and *p* = 0.168) in either case. Total polyphenol concentrations (sum of the identified polyphenol components) for the Hungarian varieties varied between 711.44 and 892.58 µg/g, while in the imported walnuts, they ranged between 413.53 and 678.71 µg/g.

### 2.2. Fatty Acids

The fatty acid composition of walnuts was measured with the GC-FID technique (Section 3.2.2) and the results are shown in Table 1. In each sample, more than 90% of the total fatty acid content was unsaturated. These results are in agreement with the studies of Kafkas et al. [32] and Liu et al. [33]. Monounsaturated fatty acid (MUFA) content ranged between 12.43% (Chandler) and 23.04% (in ‘A117’ Berzék); polyunsaturated fatty acids (PUFAs) varied between 68.9% (in ‘A117’ Pálháza) and 79.9% (in Chandler). Based on our data, the Hungarian varieties are richer in MUFAs than other walnut samples, however the imported walnuts contained more PUFA components. The saturated fatty acid (SFA) content did not exceed 9% of total fatty acids in either sample. The fatty acid profiles consisted of eight fatty acids, which all presented in the examined walnut samples. Linoleic acid (PUFA), which is an omega-6 fatty acid, plays a special role in the support of heart health, and was the most abundant in all varieties, ranging between 56.8% (in ‘A117’ Berzék) and 63.8% (in Chernivets’ky 1). The imported samples contained a significantly (*p* = 0.001) higher amount of linoleic acid than the domestic varieties. For the Hungarian walnuts, samples from Pálháza showed more favorable—higher—values, however the difference was not significant (*p* = 0.33). ‘M10’ and ‘A117’ varieties had almost the same amount of linoleic acid; the difference is only 2.3%. Oleic acid (C18:1 n-9c, one of the MUFAs) is the second most commonly detected fatty acid. The observed Hungarian varieties were rich in oleic acid because they contained almost twice as much as the other imported walnut samples, which meant a significant difference (*p* < 0.001). In comparison with other studies, Hungarian varieties also contained more oleic acid than Romanian [16] and other American [32] walnuts. Comparing the Hungarian varieties, ‘A117’ had an almost 9% higher oleic acid content than ‘M10’. ɣ-linolenic acid (GLA, C18:3 n-3c, one of the PUFAs) was the third in order of importance, ranging from 10.5% (in ‘A117’ Pálháza) to 17.9% (Brazilian sample). The difference between the Hungarian and imported walnuts was significant (*p* = 0.01) in the case of GLA content. The other detected fatty acids were palmitic acid (5.4–6.5%), stearic acid (2.0–2.4%), arachidonic acid (0.05–0.1%), palmitoleic acid (0.03–0.1%), and gondoic acid (0.1–0.2%). Bouabdallah et al. [34] and Nogales-Bueno et al. [35] also detected linoleic acid (60.42–65.77%), oleic acid (13.21–19.94%), and linoleic acid (7.61–13%) as the three major fatty acids in walnut kernels, similar to our results. WHO experts recommend a greater than 0.45 PUFA/SFA ratio for a the “balanced diet” [36]. From this aspect, all examined walnut samples play an important role in human health, with a PUFA/SFA ratio higher than 8 (Table 1).

The highest value of PUFAs/SFAs was 10.5 in the American walnut sample and the lowest was 7.98 in M10P (Table 1). These results are considerably higher compared to the Iranian walnut genotypes [37], but lower than the English, Shinano, Oni, and Hime walnuts [13]. It is important to examine the ratio of omega-6 and omega-3 essential fatty acids, as humans’ diets have evolved where this value was approx. 1. However, with the spread of the Western diet, this ratio can reach 15–20 nowadays, which can contribute to a positive health effect. The study of Zec et al. [38] indicates that six weeks of regular walnut consumption favorably modified the omega-6/omega-3 ratio in the plasma of rats. Foods with a low ratio of the two fatty acids play an important role in the daily diet. In this regard, sea buckthorn seeds stand out, with an omega-6/omega-3 ratio of 0.9–1.3 for the six examined varieties [39]. Among the tested walnut samples, ‘Chandler’ varieties grown in Brazil (3.4), the United States of America (3.5), and Chile (3.8) were closer to the suggested value of 1. In the case of European varieties (‘M10’, ‘A117’, and Chernivets’ky 1), the ratio was in the range of 4.7–5.5. Khayata et al. [40] examined Syrian varieties and determined that the omega-6/omega-3 ratio was 3.9 for fresh samples, while it was 4.1 for stored walnut samples. PCs are linear combinations of original variables and are determined so that the first PC explains the largest part of the total variance. This means that the same PC explains correlated variables and less correlated variables by different PCs. In the present analysis, the first two PCs represented 51.55 and 24.70% of the total variance in the case of eight variables. The score plot shows 76.25% of the variance (Figure 2).

In the PCA plot with two principal components, three distinct groups were identifiable. The first group was correlated with American, Chilean, and Brazilian classes, group 2 with the ‘A117’B and ‘M10’B classes from both growing areas, and group 3 was associated with Chernivets’ ky 1. The first principal component showed a high correlation (>0.900) with oleic acid, while the second principal component showed a high correlation with gondoic acid based on the correlation matrix (Table 2).

### 2.3. Minerals

The mineral composition of the walnut samples was measured by ICP-OES analysis (Section 3.2.3). The results show significant differences (*p* < 0.05) between the walnut samples among the varieties, except for iron content (Table 3). In general, the Hungarian varieties contained higher levels of calcium, magnesium, copper, phosphorus, and sodium, while the imported walnuts had a higher potassium concentration, and iron, copper, and zinc were present in almost the same number of samples. Considering the essential macro-elements, which are important for the human body, Hungarian varieties are more valuable from a nutritional point of view than the studied imported walnuts. The ‘M10’ variety had higher contents of Na, Mg, Fe, and Cu, while higher amounts of P and Zn were measured in ‘A117’ on average. In terms of the growing area, walnuts from Berzék contained higher K, Na, and Ca contents, while values were higher or equal in the cases of Mg and Zn. Cu amount was the highest in samples from Pálháza. The calcium content was in the range of 2033–2800 mg/kg, magnesium 1700–1933 mg/kg, copper 12.00–21.33 mg/kg, iron 24.67–30.67 mg/kg, phosphorus 4066–4900 mg/kg, potassium 4066–4900 mg/kg, sodium 323.00–493.33 mg/kg, and zinc 26.67–31.33 mg/kg for the eight examined walnuts. These values were higher than the values measured in other walnuts [4]. However, it is well known that the mineral content of the walnuts depends on several factors, such as plant nutrition, soil type, pH level, and the elemental composition of the soil, which greatly influence mineral absorption by plants and heat accumulation. The varieties ‘M10’ and ‘A117’ had the highest Ca (approx. 11% and 16%, respectively), Mg (approx. 6% and 2%, respectively), and Na (approx. 22% and 3%, respectively) contents. The differences can be seen compared to American ‘Chandler’. In the Chilean sample, the highest K (approx. 10%), Cu (approx. 12%), and Zn (approx. 13%) concentrations were measured, while the amount of Fe was also higher compared to the other walnuts. In the case of K, the imported walnuts had a considerably higher content on average by approx. 15% than the Hungarian varieties. In terms of the macro-elements, the Hungarian varieties (‘M10’ and ‘A117’) stood out based on their Ca, Mg, and P contents, while in the case of K, the imported samples (especially the Chilean sample) contained a larger amount, on average around 15%. Regarding the Na content, lower values of the imported varieties were physiologically more favorable. The Chilean sample had high concentrations Cu and Zn, while the amount of Fe was the highest for the ‘Milotai 10’ variety taken from both Hungarian fruit sites. The calculated (Na + Ca)/(K + Mg) ratios of the studied walnuts are shown in Table 3. Based on these calculated values, the consumption of all examined kernels contributes to maintaining the optimal proportions of cation (Na + Ca)/(K + Mg) = 1 ratios in the body. This can greatly contribute to a shifted rate due to poor nutrition to keep it at an ideal value.

Considering the mineral content of walnuts from a nutritional point of view is important to prevent certain diseases and to maintain health. Table 4 shows the calculated minerals of each variety (based on the results and the weights of kernels (kernel weights of the examined walnut varieties: ‘Milotai 10’: 7.1g/kernel, ‘Alsószentiváni 117’: 6.0 g/kernel, ‘Chandler’: 6.4 g/kernel, ‘Chernivets’ky 1’: 6.0 g/kernel)) completed with the calculated RDI% using the general recommended daily intake Rodler, 2005 [41]. With regard to Ca, P, Mg, Cu, and Fe, the M10 variety ensures adequate input values to the greatest extent. However, in the cases of K and Zn, Chilean Chandler showed the highest RDI%. 

## 3. Materials and Methods

### 3.1. Plant Materials

Walnuts (*Juglans regia* L.) were harvested in 2022. Two different varieties, ‘Milotai 10’ (M10) and ‘Alsószentiváni 117’ (A117), from two different Hungarian areas, Berzék and Pálháza, belong to the main walnut-growing region in Hungary, were involved in the trial. The walnut orchard located in Berzék was established in 2009, and the orchard located in Pálháza in 1999. Both orchards were planted as 10 m length rows with 10 m between the rows and were not irrigated. Conditions in both orchards were good, as the average shoot length was over 60 cm. Table 5 contains the data for both fruit site conditions.

The samples were collected from bearing orchards in both sites (Berzék: 48°01′20.54″ N, 20°56′51.66″ E, 101 m above sea level; Pálháza: 48°29′32.70″ N, 21°30′04.60″ E, 220 m above sea level). The soil conditions in Berzék were loamy soil with a high lime content (pH = 7.5, total lime content in the top 120 cm layer 5%) and humus content (0.8–1.5%), with medium compactness. In Pálháza, the soil was clay soil with a high lime content (pH = 7.2, total lime content in the top 120 cm layer 3%) and humus content (1.0–1.5%). The canopies were trained to a central leader canopy with regular pruning.

The samples were collected when 50% of green husks were open. After harvesting, the green husks were removed mechanically, the walnuts were washed, and the samples dried up to an 8% moisture content. The samples were cracked manually and stored at 4 °C in the dark until analysis. The other walnut samples were commercially available from Ukrainian, American, Chilean, and Brazilian imports and were purchased from local stores in fresh form. Table 6 presents the examined samples involved in the study.

#### Properties of Walnut Varieties

Milotai 10 variety (M10): This state-approved variety derives from the selection of the local population. It ripens in the month of September and develops 25% of its nuts on lateral buds. The nut is regular, spherical, 33–36 mm in diameter, with a yellowish-brown shell, and markings or reticulations on the surface. The kernel is light-yellow and tasty. The kernel set rate is 47%. ‘Milotai 10‘ represents a standard market value regarding both in-shell and shelled walnuts. The tree is moderately vigorous and develops a hemisphere-shaped canopy [42]. Alsószentiváni 117 variety (A117): This state-approved variety derives from the selection of the local population. It ripens in the 2nd week of September. It is solely a terminal bearer. Its nuts are mid-large, 33–36 mm in diameter. The shell is slightly striated, semi-hard, and light brown. Its kernel is yellowish-brown and tastes good. The kernel set rate is 48%. The tree is highly vigorous, with a little spreading canopy [42].

Chandler variety: This is the most-grown hybrid walnut variety in the world. It is the preferred variety to plant in sites with a Mediterranean climate condition. Its budbreak is medium-early and is mostly a lateral bearer. Nuts ripen in the last weeks of September. The dried fruits have a high quality with an average shell weight of 13 g, a diameter of 28–30 mm, smooth-surfaced, and have a light shell and kernel color. The kernel set rate is 49%. The tree is moderately vigorous, partly upright in habit, but highly susceptible to *Xanthomas* disease [43,44,45]. This variety is the control in our trial, more precisely American Chandler. Chernivets’ky 1 variety: This variety was selected from the local Ukrainian walnut population. The nuts ripen in the second week of September. The dried nuts have an oval shape, 10.6 to 12.2 g weight, and are 32 to 34 mm in diameter. The shell is thin (0.9 mm) and its surface is smooth. The kernel set rate is in the range of 50.7 to 54.6%, and it is easy to crack and remove the kernel. The leaves and nuts are tolerant to *Xanthomas* disease. The tree has a medium vigor [46].

Table 7 contains the most important characteristics of the examined walnut varieties.

### 3.2. Methods

#### 3.2.1. Polyphenol Analysis

The sample preparation and HPLC analysis were performed according to the methods of Bujdosó et al. [47].

A total of 1000 g of walnut seeds with their coats was used as analytical material and 1 g samples were extracted in 10 mL methanol for 12 h in the dark at 4 °C, using an Edmund Bühler SM 30 control shaker (3000 rpm). After the extraction, the supernatant was centrifuged (Hettich Mikro 22R centrifuge, 15,000 rpm for 5 min). Afterwards, the supernatant was filtered on a 0.45 μm MILLEX^®^-HV Syringe Driven Filter Unit (SLHV 013 NL, PVDF Durapore), purchased from Millipore Co. (Bedford, MA, USA), and injected into the HPLC system. The WATERS High-Performance Liquid Chromatograph (Waters Co., 34 Maple Street, Milford, MA, USA) was equipped with an absorbance detector (2487 Dual λ), a binary HPLC pump (1525), and in-line degasser, a column thermostat (set at 40 °C), and an 717plus autosampler (set at 5 °C), and was controlled using EMPOWER TM^2^ software version 2. A KINETEX C18 2.6 μm 150 × 4.6 mm column (Phenomenex 411 Madrid Avenue, Torrance, CA 90501-1430 USA) was installed, and the gradient mobile phase was A: H_2_O:MeOH: H_3_PO_4_ = 94:5:1, B: MeOH (0–30 min: A 100–10%, 30–30.1 min: 10–100%, 30.1–31: A 100%), with a flow rate 1 mL/min. The pressure in the column was 4200 ± 10 psi at a column temperature of 30 °C. The phenolic components were monitored at a wavelength of 280 nm. The method revealed four peaks, namely cinnamic acid [CAS: 140-10-3], gallic acid [CAS: 149-91-7], rutin [CAS: 207671-50-9], and catechin [CAS: 154-23-4], which were identified with reference standards. The quantities of the individual phenolic compounds are presented in μg/g.

#### 3.2.2. Fatty Acid Profile

The sample preparation and GC-FID analysis were performed according to the method of Tormási and Abrankó [48]. A total of 4 g of grounded walnuts was homogenized with 4 g of quartz sand for fat extraction. Then 1 g of each sample was centrifuged three times in an Eppendorf tube (7000 rpm, 10 min) and the released fat was removed after each round. The fatty acid composition of walnut oil was determined according to the ISO 12966-2:2017 “Rapid Method” [49] with some modifications. Briefly, 10–15 mg of fat were dispensed into a 15 mL screw-cap centrifuge tube with 1.8 mL of isooctane and 200 µL of internal standard (1 mg/mL solution of glyceryl trinonadecanoate in chloroform). After dissolving the fat, the esterified fatty acids in the sample were methylated with 200 µL of potassium hydroxide (stirring for 1 min), and after resting (2 min), 4 mL of saturated sodium chloride solution were added to the sample and homogenized (10 s).

Samples were analyzed by the GC-FID method. An Agilent (Santa Clara, CA, USA) 6890 GC-FID system equipped with an Agilent 7683 autosampler was used. For separation, a Phenomenex (Torrance, CA, USA) Zebron ZB-FAME (60 m, 0.25 mm, and 0.20 µm) column with a cyanopropyl stationary phase and hydrogen gas (1.2 mL/min) mobile phase was used. The inlet temperature was 250 °C and the detector temperature was 260 °C. A split ratio of 50:1 and 1 µL injection volume were used. The temperature program started from 100 °C, which was kept constant for 3 min. Then, the column was heated at 20 °C min^−1^ to reach 166 °C, where it was kept for 5 min. Then, it was heated to 180 °C, at 1 °C min^−1^, and finally to 240 °C at 10 °C min^−1^, where it was kept for 3 min. Fatty acids were identified by comparing the retention times of the FAME mixture and quantified using an external four-point calibration (0, 10, 20, and 40 µg/mL) and 100 µg/mL of nonadecanoic acid (dissolved in isooctane, 1 mg/mL). In the walnut samples, eight fatty acids were detected and quantified: palmitic acid, palmitoleic acid, stearic acid, oleic acid, linoleic acid, ɣ-linoleic acid, arachidonic acid, and gondoic acid. The results are expressed in relative percentages of each fatty acid.

#### 3.2.3. Mineral Composition

The samples were prepared according to MSZ EN 13805: 2002 [50] and the measurement was performed by ICP-OES equipment based on EPA Method 6010C [51]. Walnut samples were air-dried at 60 °C and then pulverized. The powdered samples were digested in a Teflon holder using the solution mixture of 2 mL of H_2_O_2_ and 2 mL of HNO_3_. After digestion, the samples were distilled with distilled water to a total volume of 10 mL. Prior to the measurement, all samples were filtered through a streamlined filter. All the samples were analyzed by ICP–OES (PerkinElmer, model: Optima 8000 ICP-OES), using winLab32 software version 3 for the analysis. The spectrometer was equipped with a Charge-Coupled Device (CCD) array detector that measures from 160 nm to 900 nm. The introduction system was composed of a glass cyclonic spray chamber and a glass concentric (Meinhard) nebulizer. The injector tube diameter of the torch was 2.0 mm. Part of the ICP-OES was also the monochromator, which detects chemical elements separately. Sample flow was 1.50 mL/min, plasma gas flow was 15 L/min, auxiliary gas flow was 0.3 L/min, and nebulizer gas flow was 0.6 L/min. Five macro-elements, Ca, K, Mg, Na, and P, and three microelements, Cu, Fe, and Zn, were determined, and their concentrations were presented in mg/kg.

#### 3.2.4. Statistical Analysis

Mean values and standard deviations are reported. All the experiments were performed in triplicate. Statistical analysis was performed by ANOVA IBM SPSS v.27 at a 95% confidence interval with one fixed factor (variety). Superscripts with lower-case letters indicate significant differences by variety along the rows in the case of polyphenols, fatty acids, and minerals. Principal component analysis (PCA) (SPSS software v. 27) was used to compare multiple independent groups for the fatty acid profile. The variance eigenvalue was greater than 1. The loadings (or factor scores) corresponding to the principal components were calculated from the correlation matrix.

## 4. Conclusions

In this study, the observed walnut samples had different nutritional values. Hungarian varieties were bred and grown in Hungarian climatic conditions. Especially ‘Milotai 10’ had an outstanding compounds content. It had higher rutin, oleic acid, and palmitic acid, as well as calcium, magnesium, natrium, phosphorus, and iron contents. In terms of the fatty acid profile, Hungarian varieties were rich in MUFAs, especially in oleic acid, but contained less PUFAs than the other walnut samples. Comparing the fruit sites in Hungary, there was no obvious difference. ‘Chandler’ harvested from Chile contained more potassium, copper, and zinc than the other examined varieties. ‘Chernivets’ky 1’ harvested in Ukraine had outstanding cinnamic acid and linoleic acid contents than the other varieties. All ‘Chandler’ samples (mainly the American variety) contained 11% more PUFAs compared to the samples derived from Hungary. It was concluded that, overall, the Hungarian walnut varieties had high nutritional values. Therefore the labeling of the nut’s origin is important for consumers.

## Figures and Tables

**Figure 1 plants-13-02097-f001:**
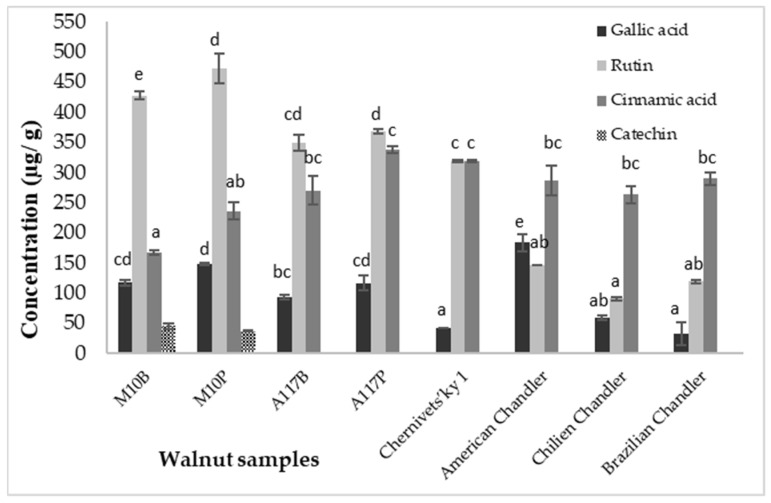
Concentration of some polyphenols in the walnut samples. A117B/P: Alsószentiváni 117 from Berzék/Pálháza; M10B/P: Milotai10 from Berzék/Pálháza; ^a,b,c,d,e^: different letters indicate significant statistical differences (Tukey’s test, *p* < 0.05).

**Figure 2 plants-13-02097-f002:**
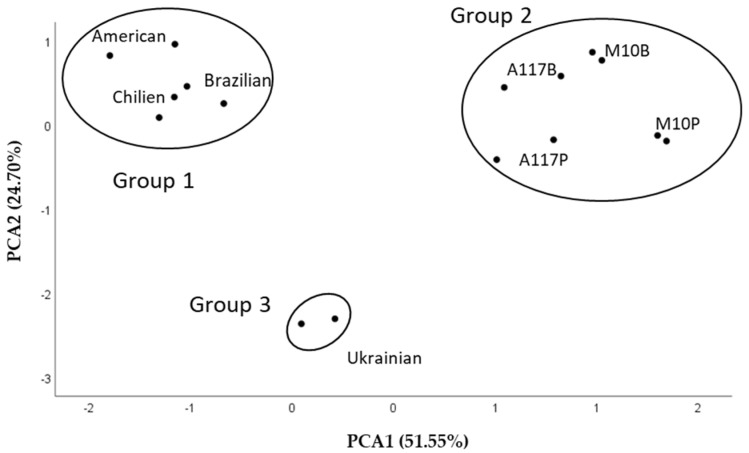
Score plot of PCA for fatty acids. A117B/P: Alsószentiváni 117 from Berzék/Pálháza; M10B/P: Milotai10 from Berzék/Pálháza.

**Table 1 plants-13-02097-t001:** Fatty acid composition of studied walnut samples.

Walnut Samples								
m/m%	M10B	M10P	A117B	A117P	Chernivets’ky 1	American Chandler	Chilien Chandler	Brazilian Chandler
C16:0	6.1 ± 0.02 ^e^	6.5 ± 0.00 ^f^	5.6 ± 0.01 ^b^	5.7 ± 0.01 ^bc^	5.9 ± 0.01 ^d^	5.4 ± 0.01 ^ab^	5.6 ± 0.01 ^b^	5.5 ± 0.08 ^b^
C18:0	2.0 ± 0.05 ^ad^	2.3 ± 0.02 ^cd^	2.3 ± 0.01 ^ab^	2.3 ± 0.02 ^cd^	2.4 ± 0.03 ^e^	2.1 ± 0.05 ^a^	2.0 ± 0.01 ^a^	2.2 ± 0.01 ^bc^
* C20:0	0.1 ± 0.01	0.1 ± 0.01	0.1 ± 0.01	0.1 ± 0.01	0.05 ± 0.01	0.1 ± 0.02	0.05 ± 0.01	0.1 ± 0.01
C16:1n-7c	0.1 ± 0.00 ^c^	0.1 ± 0.03 ^c^	0.04 ± 0.01 ^b^	0.04 ± 0.01 ^ab^	0.03 ± 0.01 ^a^	0.03 ± 0.00 ^a^	0.03 ± 0.01 ^a^	0.03 ± 0.00 ^a^
C18:1n-9c	21.6 ± 0.03 ^f^	20.0 ± 0.04 ^e^	22.8 ± 0.07 ^g^	22.7 ± 0.02 ^g^	15.3 ± 0.02 ^a^	12.2 ± 0.01 ^a^	14.1 ± 0.02 ^b^	13.3 ± 0.11 ^ab^
* C20:1n-9c	0.2 ± 0.02	0.2 ± 0.01	0.2 ± 0.01	0.2 ± 0.01	0.1 ± 0.02	0.2 ± 0.03	0.2 ± 0.04	0.2 ± 0.02
C18:2n-6c	58.5 ± 0.1 ^b^	59.5 ±0.1 ^c^	56.8 ±0.1 ^a^	58.4 ± 0.0 ^ab^	63.8 ± 0.1 ^f^	62.3 ± 0.13 ^eg^	62.1 ± 0.1 ^eg^	60.8 ± 0.32 ^ag^
C18:3n-3c	11.4 ± 0.01 ^bg^	11.5 ± 0.01 ^b^	12.2 ± 0.04 ^c^	10.5 ± 0.00 ^a^	12.3 ± 0.01 ^c^	17.6 ± 0.02 ^f^	16.0 ± 0.03 ^d^	17.9 ± 0.14 ^f^
SFA	8.2 ± 0.08	8.9 ± 0.05	8.0 ± 0.05	8.1 ± 0.04	8.4 ± 0.05	7.6 ± 0.08	7.7 ± 0.05	7.8 ± 0.10
MUFA	21.9 ± 0.03	20.3 ± 0.04	23.04 ± 0.07	22.94 ± 0.02	15.43 ± 0.02	12.43 ± 0.01	14.33 ± 0.02	13.53 ± 0.11
PUFA	69.9 ± 0.11	71.0 ± 0.11	69.0 ± 0.05	68.9 ± 0.01	76.1 ± 0.02	79.9 ± 0.15	78.1 ± 0.13	78.7 ± 0.46
PUFA/SFA	8.52	7.98	8.63	8.51	9.06	10.51	10.14	10.09
ɷ-6/ɷ-3	5.1	5.2	4.7	5.5	5.2	3.5	3.8	3.4

A117B/P: Alsószentiváni 117 from Berzék/Pálháza; M10B/P: Milotai10 from Berzék/Pálháza; ^a,b,c,d,e,f,g^: different letters indicate significant statistical differences (Tukey’s test, *p* < 0.05). * Means there are no significant differences.

**Table 2 plants-13-02097-t002:** Correlation values of fatty acids with principal components.

Fatty Acids	PCA1	PCA2
Palmitic acid	0.752	−0.234
Stearic acid	0.318	−0.687
Arachinodic acid	0.690	0.370
Palmitoleic acid	0.871	0.276
Oleic acid	0.911	0.071
Gondoic acid	−0.250	0.940
Linoleic acid	−0.739	−0.486
ɣ-linolenic acid	−0.885	0.328

**Table 3 plants-13-02097-t003:** Macro- and micro-element contents of walnut samples.

Walnut Varieties								
(mg/kg)	M10B	M10P	A117B	A117P	Chernivets’ky 1	American Chandler	Chilean Chandler	Brazilian Chandler
Macro-elements								
Ca	2800 ± 100 ^b^	2400 ± 100 ^abc^	2800 ± 200 ^b^	2666 ± 230 ^bc^	2033 ± 57 ^a^	2333 ± 208 ^ab^	2600 ± 20 ^bc^	2333 ± 115 ^ab^
K	4833 ± 58 ^ab^	4500 ± 0 ^a^	4800 ± 173 ^ab^	4733 ± 850 ^a^	5433 ± 289 ^ab^	5333 ± 551 ^ab^	5866 ± 404 ^b^	5100 ± 200 ^ab^
Mg	1933 ± 58 ^b^	1900 ± 265 ^b^	1833 ± 58 ^ab^	1833 ± 252 ^ab^	1700 ± 100 ^bc^	1800 ± 153 ^bc^	1800 ± 100 ^bc^	1800 ± 0 ^bc^
Na	493.33 ± 29 ^ab^	400 ± 26 ^ab^	453 ± 104 ^ab^	337 ± 47 ^a^	323.00 ± 100.00 ^a^	370.00 ± 26.46 ^ab^	360.00 ± 40.00 ^ab^	370.00 ± 34.64 ^ab^
P	4766 ± 115 ^bcd^	4233± 58 ^abcd^	4900 ± 100 ^cd^	4933 ± 416 ^d^	4233 ± 115 ^ab^	4167 ± 404 ^ab^	4266 ± 208 ^abc^	4066 ± 153 ^a^
Micro-elements								
Cu	18.33 ± 1.15 ^b^	20.00 ± 0.00 ^b^	17.33 ± 0.58 ^b^	19.00 ± 2.65 ^b^	12.00 ± 1.00 ^a^	19.00 ± 2.00 ^b^	21.33 ± 2.08 ^b^	18.33 ± 1.15 ^b^
Fe	30.67 ± 2.08 ^ab^	28.33 ± 1.53 ^b^	24.67 ± 1.15 ^bc^	25.33 ± 4.51 ^bc^	25.00 ± 1.00 ^b^	27.00 ± 4.00 ^b^	28.33 ± 0.58 ^b^	27.33 ± 3.51 ^b^
Zn	30.00 ± 1.00 ^ab^	28.00 ± 1.00 ^a^	31.33 ± 0.58 ^ab^	31.33 ± 4.16 ^ab^	26.67 ± 2.08 ^a^	31.00 ± 3.00 ^ab^	35.01 ± 3.00 ^b^	27.12 ± 1.00 ^a^
(Na + Ca)/(K + Mg)	0.38 ± 0.01	0.39 ± 0.01	0.49 ± 0.02	0.56 ± 0.20	0.39 ± 0.03	0.48 ± 0.03	0.33 ± 0.02	0.46 ± 0.05

A117B/P: Alsószentiváni 117 from Berzék/Pálháza; M10B/P: Milotai10 from Berzék/Pálháza; ^a,b,c,d^: different letters indicate significant statistical differences (Tukey’s test, *p* < 0.05).

**Table 4 plants-13-02097-t004:** Calculated mineral content (mg) and the percentage of recommended daily intake (%) of different walnut varieties.

Minerals	Recommended Daily Intake (mg/Day) [41]	mg Minerals/1 pcs Kernel (% RDI/1 pcs. Kernel)							
		‘M10B’	‘M10P’	‘A117B’	‘A117P’	‘Chernivets’ky 1’	‘American Chandler’	‘Chilean Chandler’	‘Brazilian Chandler’
K	3500	34.31 (1.0)	31.95 (0.9)	28.80 (0.8)	28.44 (0.8)	32.59 (0.9)	34.13 (0.9)	37.54 (3.5)	32.64 (0.9)
Ca	800	19.88 (2.5)	17.04 (2.1)	16.80 (2.1)	15.99 (1.9)	12.19 (1.5)	14.93 (1.8)	16.64 (2.1)	14.93 (1.8)
P	620	34.31 (1.0)	30.05 (4.8)	29.40 (4.7)	29.60 (4.8)	25.39 (4.1)	26.67 (4.3)	27.30 (4.4)	26.02 (4.2)
Mg	350	13.72 (3.9)	13.49 (3.8)	10.99 (3.1)	10.99 (3.1)	10.20 (2.9)	11.52 (3.3)	11.52 (3.3)	11.52 (3.3)
Cu	1.1	0.13 (11.8)	0.14 (12.9)	0.10 (9.4)	0.11 (10.3)	0.07 (6.5)	0.12 (11.1)	0.13 (12.4)	0.12 (10.6)
Zn	9	0.21 (2.3)	0.19 (2.2)	0.18 (2.1)	0.18 (2.1)	0.16 (1.8)	0.19 (2.2)	0.22 (2.5)	0.17 (1.9)
Fe	12.5	0.22 (1.7)	0.20 (1.6)	0.14 (1.2)	0.15 (1.7)	0.16 (1.7)	0.17 (1.4)	0.18 (1.4)	0.17 (1.4)

A117B/P: Alsószentiváni 117 from Berzék/Pálháza; M10B/P: Milotai10 from Berzék/Pálháza.

**Table 5 plants-13-02097-t005:** Fruit site conditions of walnut orchards located in Berzék (Hungary) and in Pálháza (Hungary) in 2022.

Category	Berzék	Pálháza
average yearly temperature	11.4 °C	10.9 °C
average yearly temperature during the growing season (between April and September)	18.4 °C	17.9 °C
average minimum temperature during spring (between March and May)	2.4 °C	1.9 °C
number of days with frost during spring (between March and May)	36	40
average yearly luminous flux	1032 L/m^2^/day	1005 L/m^2^/day
average yearly precipitation	443.2 mm	468.2 mm

**Table 6 plants-13-02097-t006:** The examined walnut varieties from different growing areas.

	Varieties			
Growing Areas	‘Milotai 10’	‘Alsószentiváni 117’	‘Chandler’	‘Chernivets’ky 1’
Berzék, Hungary	×	×		
Pálháza, Hungary	×	×		
United States of America			×	
Chile			×	
Brazil			×	
Ukraine				×

**Table 7 plants-13-02097-t007:** Main characters of the examined walnut varieties.

Varieties	Harvest Time	Dried Nut Weight (g)	Kernel Percentage (%)	Kernel Weight (g)
‘Milotai 10’	3rd week of September	15	47	7.1
‘Alsószentiváni 117’	2nd week of September	12.5	48	6
‘Chandler’	3rd week of September	13.0	49	6.4
‘Chernivets’ky 1’	2nd week of September	11.4	53	6.0

## Data Availability

Data are contained within the article.

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
