# Peer review of "Comparative Study of Different Walnut (Juglans regia L.) Varieties Based on Their Nutritional Values"

_plants, 2024, doi:10.3390/plants13152097_

Round 1
Reviewer 1 Report
Comments and Suggestions for Authors
Author compared different walnuts varieties based on different nutritional values. Results shown that Milotai 10’ is an outstanding walnut due to high nutritional properties. and, if grown in Carpathian basin, might be a promising one for raw consumption and industrial purposes.
The article is well written and data presentation is excellent however i am suggesting some minor changes prior to the acceptance of this work.
1. Article title need to re-phrase and avoid capital letter for each word, except the initial word of the sentence and the species name.
2. The first para of introduction is too short, please complete it.
3. Text in the introduction section is not enought, also need to provide more literture about different varieties nutrients contents.
4. Author highlights the aim of work but not explain the hypothesis/novelty statements, please add.
5. Line 74, is grammatically wrong, please carefully check, such as, The concentration of..., and for different phenols must used (are) not is shown.
6. I suggest to arrange the results text, such as which walnut variety was best in which region and which showed decreased in the nutrients, and show the results in %. Thus, readers will understand your findings, the current format of results are confusing.
7. For Table 1, the end row is missing, please check.
8. Figure 2, labels need to revise, such as, first and second principal components need to replace with PCA1 and PCA2.
9. In the trial (line 218)
10. Must use a Table for different regions and different varieties of walnuts, for easy understanding in the materials and methods sections.
11. I suggest to add more literature for the support of your results.
Author Response
Reviewer 1
list of point to point changes
Dear Reviewer1,
first of all, I like to take the opportunity to say Thank you for your time and effords to read, review and correct our paper. Here you can see our answers of your questions.
Yours sincerely,
the authors
Comments 1. Article title need to re-phrase and avoid capital letter for each word, except the initial word of the sentence and the species name.
Response 1: We made the changes requested.
Comments 2. The first para of introduction is too short, please complete it.
Response 2: The authors completed the first para.
Comments 3. Text in the introduction section is not enought, also need to provide more literture about different varieties nutrients contents.
Response 3: The authors completed the text in the introduction.
Comments 4. Author highlights the aim of work but not explain the hypothesis/novelty statements, please add.
Response 4: Novelty value:
Line 68. Aim of this research work was to characterize and evaluate several commercially available walnut varieties on the Hungarian market, which is the first study in this area in Hungary. Line 73. Our hypothesis is, that the fruit site conditions have some effects on the compounds, can be detected in the kernel.
Comments 5. Line 74, is grammatically wrong, please carefully check, such as, The concentration of..., and for different phenols must used (are) not is shown.
Response 5: The sentence was corrected, such as „The concentration of cinnamic acid, gallic acid, rutin and catechin is shown in Figure 1. In case of Hungarian varieties, rutin was present in the highest quantity among the examined polyphenols.”
Comments 6. I suggest to arrange the results text, such as which walnut variety was best in which region and which showed decreased in the nutrients, and show the results in %. Thus, readers will understand your findings, the current format of results are confusing.
Response 6: Thank you for this comment. We added these additional information.
Comments 7. For Table 1, the end row is missing, please check.
Response 7: This Table was checked.
Comments 8. Figure 2, labels need to revise, such as, first and second principal components need to replace with PCA1 and PCA2.
Response 8: The authors corrected the Figure 2.
Comments 9. In the trial (line 218)
Response 9: This was corected.
Comments 10. Must use a Table for different regions and different varieties of walnuts, for easy understanding in the materials and methods sections.
Response 10: A new Table (Table 6) about this issue was inserted.
Comments 11. I suggest to add more literature for the support of your results.
Response 11: More references was added to the manuscript.
Reviewer 2 Report
Comments and Suggestions for Authors
Walnuts are a health food. Varieties, origin, processing methods, and storage time all directly and significantly affect the nutritional composition of walnuts. This manuscript investigates the contents of phenolics, fatty acids, and minerals in 8 walnuts on the Hungarian market and draws the corresponding conclusions. The main issues with this paper are:
1. The current manuscript only analyzes 8 walnut samples, the sample size is small, and the representativeness is low, insufficient to support the relevant statements in "4 Conclusion" Was there market data supporting these 8 walnuts as the main walnut varieties and production areas in the Hungarian market?
2. The introduction states that China and the United States are the largest walnut producers, but the manuscript only examined one American variety and no Chinese varieties.
3. In addition to phenolics, fatty acid, and mineral content, amino acids and active peptides content, and flavor composition are also important quality indicators of walnuts. The study needs to test more indicators.
4. Processing methods and storage time directly affect the nutritional composition of walnuts. The 8 samples in the manuscript should have consistency.
5. Line 38, "L." should not be italicized.
6. Line 142, this should be "The highest value of PUFA/SFA was 10.51 in the American walnut sample and the lowest was 7.98 in M10P."
Author Response
Reviewer 2
list of point to point changes
Dear Reviewer2,
Many Thanks for having time and efforts to review our paper. All of your questions, comments, and reactions are very useful for us. Here you can see our answers.
Yours sincerely,
the authors
Comments 1. The current manuscript only analyzes 8 walnut samples, the sample size is small, and the representativeness is low, insufficient to support the relevant statements in "4 Conclusion" Was there market data supporting these 8 walnuts as the main walnut varieties and production areas in the Hungarian market?
Response 1: The examined varieties play a dominant role on the Hungarian market. There aren’t any more varieties on shelves of the supermarkets.
Comments 2. The introduction states that China and the United States are the largest walnut producers, but the manuscript only examined one American variety and no Chinese varieties.
Response 2: Varieties derived from China are not in the Hungarian production. At the moment, there are any walnuts, grown in China, exported to Hungary.
Comments 3. In addition to phenolics, fatty acid, and mineral content, amino acids and active peptides content, and flavor composition are also important quality indicators of walnuts. The study needs to test more indicators.
Response 3: Our manuscript is a prliminary study. Many Thanks for your suggestion, in the near future, we are using them. In the current manuscript, we were focusing on the essential compounds derived from kernel.
Comments 4. Processing methods and storage time directly affect the nutritional composition of walnuts. The 8 samples in the manuscript should have consistency.
Response 4: Thank you for your idea. We didn’t focus on that issues, you suggested. We examined the walnut samples, in which status the final consumers can purchase them.
Comments 5. Line 38, "L." should not be italicized.
Response 5: We corrected it.
Comments 6. Line 142, this should be "The highest value of PUFA/SFA was 10.51 in the American walnut sample and the lowest was 7.98 in M10P."
Response 6: We modified it.
Reviewer 3 Report
Comments and Suggestions for Authors
The paper “Comparative Study of Different Walnut (Juglans regia L.) Varieties Based on Their Nutritional Values” analyzes eight walnut varieties as regards their nutritional values. Some polyphenols (cinnamic and gallic acids, rutin and catechin), some fatty acids (palmitic, stearic, arachidic, palmitoleic, oleic, eicosenoic (gondoic), linoleic, gamma-linoleic and alfa-linolenic acids) and some mineral substances (sodium, potassium, magnesium, calcium, iron, copper, zinc, and phosphorus) were found and quantitatively determined in the walnut samples.
The study is properly designed, the methodology is adequate, and the results are relevant and interesting. However, you should rephrase a lot of sentences. Although I am not qualified to assess the quality of English in this paper, I appreciate that an extensive editing of English language is required.
Lines 25–27: In comparison, Hungarian varieties grown in Hungarian climate conditions had on average approximately 33% more polyphenols, approximately 22% more SFA, and approximately 27% more MUFA.
Lines 35–36: Origin of different food products is compulsory to be marked on them.
Lines 68–69: ... were compared as regards their nutritional values
Line 84: ‘A117’ had an almost 15% higher cinnamic acid concentration than ‘M10’ in both growing areas.
Lines 107–109: Monounsaturated fatty acids (MUFA) content ranged between 12.43% (in American Chandler) and 23.04% (in ‘A117’ Berzék), polyunsaturated fatty acids (PUFA) varied between 68.9% (in ‘A117’ Pálháza) and 79.9% (in American variety).
Line 114–116: Linoleic acid (LA, C18:2 n-6c, one of the PUFAs), which is an omega-6 fatty acid and plays a special role in heart health, was the most abundant one in all varieties, and ranged between 56.8% (in ‘A117’ Berzék) and 63.8% (in Chernivetsky’1).
Lines 120–121: Oleic acid (C18:1 n-9c, one of the MUFAs)
Lines 125–126: Comparing the Hungarian varieties, ‘A117’ had an almost 9% higher oleic acid content than ‘M10’.
Lines 126: The ɣ-linolenic acid (GLA, C18:3 n-3c, one of the PUFAs)
Lines 141–142: 10.51 ……………. 8.51
Lines 148–149: The study of Zec et al. [26] indicates that 6 weeks of regular walnut consumption favourably modified the omega-6/omega-3 ratio in the plasma of rats.
Lines 177–179, 352–353: In general, the Hungarian varieties contained higher level of calcium, magnesium, copper, phosphorus, sodium while the import walnuts had higher potassium concentration and the iron, copper, and zinc were present almost the same amount of the samples.
Keywords: word ,,compounds" should be replaced with ,,Juglans regia L."
Line 73, 269: polyphenols, not polyhenols
Comments on the Quality of English LanguageYou should rephrase a lot of sentences.
Although I am not qualified to assess the quality of English in this paper, I appreciate that an extensive editing of English language is required.
Author Response
Reviewer 3
list of point to point changes
Dear Reviewer3,
the authors like to take the opportunity to thank your time for reviewing this paper. All of your suggestions are very useful for us. Here you can see our answers.
Yours sincerely,
the authors
Comments and Suggestions for Authors
Comments 1: The paper “Comparative Study of Different Walnut (Juglans regia L.) Varieties Based on Their Nutritional Values” analyzes eight walnut varieties as regards their nutritional values. Some polyphenols (cinnamic and gallic acids, rutin and catechin), some fatty acids (palmitic, stearic, arachidic, palmitoleic, oleic, eicosenoic (gondoic), linoleic, gamma-linoleic and alfa-linolenic acids) and some mineral substances (sodium, potassium, magnesium, calcium, iron, copper, zinc, and phosphorus) were found and quantitatively determined in the walnut samples.
The study is properly designed, the methodology is adequate, and the results are relevant and interesting. However, you should rephrase a lot of sentences. Although I am not qualified to assess the quality of English in this paper, I appreciate that an extensive editing of English language is required.
Response 1: All changes were made yellow colour in the text.
Lines 25–27: In comparison, Hungarian varieties grown in Hungarian climate conditions had on average approximately 33% more polyphenols, approximately 22% more SFA, and approximately 27% more MUFA.
Lines 35–36: Origin of different food products is compulsory to be marked on them.
Lines 68–69: ... were compared as regards their nutritional values
Line 84: ‘A117’ had an almost 15% higher cinnamic acid concentration than ‘M10’ in both growing areas.
Lines 107–109: Monounsaturated fatty acids (MUFA) content ranged between 12.43% (in American Chandler) and 23.04% (in ‘A117’ Berzék), polyunsaturated fatty acids (PUFA) varied between 68.9% (in ‘A117’ Pálháza) and 79.9% (in American variety).
Line 114–116: Linoleic acid (LA, C18:2 n-6c, one of the PUFAs), which is an omega-6 fatty acid and plays a special role in heart health, was the most abundant one in all varieties, and ranged between 56.8% (in ‘A117’ Berzék) and 63.8% (in Chernivetsky’1).
Lines 120–121: Oleic acid (C18:1 n-9c, one of the MUFAs)
Lines 125–126: Comparing the Hungarian varieties, ‘A117’ had an almost 9% higher oleic acid content than ‘M10’.
Lines 126: The ɣ-linolenic acid (GLA, C18:3 n-3c, one of the PUFAs)
Lines 141–142: 10.51 ……………. 8.51
Lines 148–149: The study of Zec et al. [26] indicates that 6 weeks of regular walnut consumption favourably modified the omega-6/omega-3 ratio in the plasma of rats.
Lines 177–179, 352–353: In general, the Hungarian varieties contained higher level of calcium, magnesium, copper, phosphorus, sodium while the import walnuts had higher potassium concentration and the iron, copper, and zinc were present almost the same amount of the samples.
Comments 2: Keywords: word ,,compounds" should be replaced with ,,Juglans regia L."
Response 2: Accepted.
Comments 3: Line 73, 269: polyphenols, not polyhenols
Response 3: Accepted.
Comments 4: Comments on the Quality of English Language
You should rephrase a lot of sentences.
Although I am not qualified to assess the quality of English in this paper, I appreciate that an extensive editing of English language is required.
Response 4: Many Thanks for your time and all of your effords. We modified all remarks, made by you.
Reviewer 4 Report
Comments and Suggestions for Authors
The title - to add the studied medicinal product: walnut In the abstract you wrote that you have
SFA and MUFA. Specify the names for these abbreviations. Also, please give a little more explanation for the 8 walnut samples taken in the study, so that it is easier to understand what they are ("eight Persian walnut (Juglans regia L.) samples: 'Milotai 10' and 'Alsószentiváni 117' Hungarian varieties derived from two Hungarian growing areas, 'Chernivets'ky 1' from Ukraine and 'Chandler' from United States of America, Chile and Brazil purchased on the Hungarian consumer market") or use abbreviations.
In Introduction: to be completed with more phytochemical and pharmacological data of walnuts.
To create a table with the 8 walnut samples studied and some specific characteristics (possibly under Material) For each sub-chapter of the
"2. Results and discussion" section, insert a sentence with information about what was followed, how it was done, by what method, etc. Line 302 corrected: GC-FID Method
It would be necessary if the results obtained per nut could be presented, especially from a nutritional point of view.
It would be necessary if the results obtained per walnut could be presented, especially from a nutritional point of view. For the prevention of some diseases and for the preservation of health, could you recommend the optimal amount of walnut kernels (a daily requirement)?
Specify the novelty and innovation of this study.
Author Response
Reviewer 4
list of point to point changes
Dear Reviewer4,
the authors are very grateful for your time to review our paper. All of your comments are very useful for us. Here you can see our answers.
Yours sincerely,
the authors
Comments and Suggestions for Authors
Comments 1: The title - to add the studied medicinal product: walnut In the abstract you wrote that you have SFA and MUFA. Specify the names for these abbreviations. Also, please give a little more explanation for the 8 walnut samples taken in the study, so that it is easier to understand what they are ("eight Persian walnut (Juglans regia L.) samples: 'Milotai 10' and 'Alsószentiváni 117' Hungarian varieties derived from two Hungarian growing areas, 'Chernivets'ky 1' from Ukraine and 'Chandler' from United States of America, Chile and Brazil purchased on the Hungarian consumer market") or use abbreviations.
Response 1: We added a new Table about origin of examined walnut varieties (Table 5) to the manuscript. And we added explanations to the abbreviations used in the abstract.
Comments 2: In Introduction: to be completed with more phytochemical and pharmacological data of walnuts.
Response 2: The introduction was completed with some new information.
Comments 3: To create a table with the 8 walnut samples studied and some specific characteristics (possibly under Material)
Response 3: The requested Table (Table 6) was added.
Comments 4: For each sub-chapter of the "2. Results and discussion" section, insert a sentence with information about what was followed, how it was done, by what method, etc. Line 302 corrected: GC-FID Method
Response 4: It was corrected.
Comments 5: It would be necessary if the results obtained per nut could be presented, especially from a nutritional point of view.
Response 5: These information were added.
Comments 6: It would be necessary if the results obtained per walnut could be presented, especially from a nutritional point of view. For the prevention of some diseases and for the preservation of health, could you recommend the optimal amount of walnut kernels (a daily requirement)?
Specify the novelty and innovation of this study.
Response 6: These information were added.
Round 2
Reviewer 2 Report
Comments and Suggestions for Authors
Thank you for your response and the revisions to the manuscript. Unfortunately, I still personally believe that the current manuscript is not suitable for publication as a research paper in the journal Plants.
1. The manuscript currently contains insufficient research content and needs additional data.
2. There are errors in the significance tests for rutin in Figure 1 and Ca in Table 3, which need to be corrected. The authors should also check the corresponding results in other data. Additionally, the significance test results for SFA, PUFA in Table 1, and (Na+Ca)/(K+Mg) in Table 3 need to be marked accordingly.
3. In my opinion, concluding that " It was concluded that overall, the Hungarian walnut varieties had a high nutritional value" is inappropriate without clear information on other walnut varieties and comprehensive investigation.
Author Response
Dear Reviewer2,
Many Thanks for your comments to improve the quality of the paper. Here you can see our answers.
Yours sincerely,
the authors
Comments 1. The manuscript currently contains insufficient research content and needs additional data.
Response 1: This paper is about a preliminary study, the most important compounds were checked. The authors think that there results are interesting and this study can be a good start of a larger one. We also believe, that for a larger study we need to find some partners. To find some partners, we need to show our results.
Comments 2. There are errors in the significance tests for rutin in Figure 1 and Ca in Table 3, which need to be corrected. The authors should also check the corresponding results in other data. Additionally, the significance test results for SFA, PUFA in Table 1, and (Na+Ca)/(K+Mg) in Table 3 need to be marked accordingly.
Response 2: The Figure 1 and the Table 3 were corrected. Values of SFA, PUFA ,MUFA and the PUFA/SFA and omega6/omega 3 ratio are calculated data based on the different values of the samples. We can not calculate unfortunately the significance letters now.
Comment 3. In my opinion, concluding that " It was concluded that overall, the Hungarian walnut varieties had a high nutritional value" is inappropriate without clear information on other walnut varieties and comprehensive investigation.
Response 3: The authors completed the cited sentence, here you can find the completed version “It was concluded that overall, the Hungarian-bred walnut varieties, grown among Hungarian climate conditions, had a high nutritional value compared to the US-bred 'Chandler', cultivated in Chile, Brazil, and the United States of America, as well as Ukrainian-bred 'Chernivetsky 1', harvetested in Ukraine.”